# Association of Body Composition with Type 2 Diabetes: A Retrospective Chart Review Study

**DOI:** 10.3390/ijerph18094421

**Published:** 2021-04-21

**Authors:** Chia-Ling Lin, Neng-Chun Yu, Hsueh-Ching Wu, Yung-Yen Lee, Wan-Chun Lin, I-Ying Chiu, Wu-Chien Chien, Yuan-Ching Liu

**Affiliations:** 1Department of Nursing, Chang Gung University of Science and Technology, Taoyuan 33303, Taiwan; lcling@mail.cgust.edu.tw; 2Neng-Chun Diabetes Clinic, Yilan County 265, Taiwan; dm9556670@gmail.com (N.-C.Y.); giawamusi@gmail.com (Y.-Y.L.); mich3226@hotmail.com.tw (W.-C.L.); yvanne1009@gmail.com (I.-Y.C.); 3Department of Nursing, Hsin Sheng Junior College of Medical Care and Management, Taoyuan 33303, Taiwan; whc06082002@yahoo.com.tw; 4Department of Medical Research, Tri-Service General Hospital, National Defense Medical Center, Taipei 11490, Taiwan; 5School of Public Health, National Defense Medical Center, Taipei 11490, Taiwan; 6Graduate Institute of Life Science, National Defense Medical Center, Taipei 11490, Taiwan

**Keywords:** body composition, body fat mass, body mass index, type 2 diabetes, skeletal muscle mass

## Abstract

This study analyzed the body composition of individuals with type 2 diabetes (T2DM). In this retrospective chart review study, body composition was measured through multifrequency bioelectrical impedance analysis (InBody 770). Body composition assessments were conducted in individuals with T2DM, who were aged ≥18 years. The parameters included body mass index (BMI), body fat mass (BFM), fat-free mass (FFM), visceral fat area, percent body fat (PBF), appendicular skeletal muscle mass (ASM), and skeletal muscle index (SMI). One-way ANOVA and independent *t*-tests were used to calculate differences in body composition distribution by age and sex. A total of 2404 participants were recruited. The prevalence of overall low muscle mass and sarcopenic obesity was 28.0% and 18.7%, respectively, which increased with age. The overall prevalence of obesity when PBF was used was 71.5%, which was higher than that when BMI was applied (32.4%). The normal BMI group exhibited a prevalence of low muscle mass of 55.6% and sarcopenic obesity of 34.8%. For both men and women, bodyweight, BFM, FFM, ASM, and SMI all decreased with age. The prevalence of low muscle mass and sarcopenic obesity was high in older adults and people with normal BMI. Using BMI to assess obesity and determine insufficient muscle mass underestimates the prevalence of obesity and neglects the problems of sarcopenia and high body fat in people with normal BMI.

## 1. Introduction

Type 2 diabetes mellitus (T2DM) is among the most severe and urgent health problems worldwide. The International Diabetes Federation estimates that the T2DM prevalence among adults aged 20 to 79 years will increase from 9.3% in 2019 to 10.2% in 2030. Specifically, the number of individuals with T2DM will increase from 463 million in 2019 to 578 million in 2030. Global medical expenses related to diabetes are estimated to reach USD 727 billion [1,2].

T2DM is closely correlated with obesity, which is commonly measured using body mass index (BMI). However, BMI cannot distinguish muscle mass and fat mass and does not consider fat distribution in the body, which are crucial limitations. Studies have indicated that T2DM is more closely correlated with body fat percentage and skeletal muscle than BMI [3,4].

Body composition is the ratio of fat to nonfat (bone, muscle, and other tissue) in body weight [5] and is often used as a crucial reference for assessing personal health, nutritional status, motor ability, and physical fitness. In addition, it is closely correlated with pathogenesis [6]. Body composition includes BMI, fat mass, percentage body fat (PBF), fat-free mass (FFM), muscle mass, and bone mass [7]. Studies have verified that changes in specific components of body composition increase mortality [8,9] and cardiometabolic risk [10,11].

BMI provides a fast and convenient standard for evaluating obesity. However, research has indicated that differences in body composition, such as differences in fat mass and fat distribution, have varying effects on health [12,13]. Abdominal obesity increases the risk of chronic diseases, such as cardiovascular diseases, hypertension, and T2DM. Excessive fat tissue has been determined to release inflammatory substances, such as interleukin-6 (IL-6). These substances have adverse effects on endothelial cells, causing higher insulin resistance and developing diabetes and dyslipidemia [3,14,15]. Fat accumulates in the entire body. Fat in the abdominal organs produces fatty acids and other substances that promote the release of inflammatory substances in the blood, blocking insulin metabolism in the liver and insulin sensitivity in the surrounding tissue. This leads to increased insulin resistance, which prevents limits blood sugar control [11]. High visceral fat levels are associated with developing metabolic syndrome [13] and hyperuricemia in nonobese adults [16]. Consequently, BMI is not the only index used for obesity assessment; body fat content and distribution are also used.

T2DM has been reported to be associated with an increased risk of sarcopenia, and it possibly contributes to disability and metabolic problems in elderly adults [17,18]. One study indicated a bidirectional relationship between T2DM and sarcopenia [19]. Furthermore, compared with sarcopenia or obesity alone, sarcopenic obesity has a much greater health impact and may have a synergistic effect with low-grade inflammation and thus exacerbate insulin resistance, further impairing glucose metabolism [20]. Therefore, this study analyzed the association of body composition with T2DM through a retrospective chart review.

## 2. Materials and Methods

### 2.1. Study Design and Participants

This was a retrospective chart review study. A retrospective observation survey was conducted on 2404 patients who visited a diabetes clinic in Northern Taiwan more than once. The inclusion criteria were age older than 18 years and completion of body composition assessments. The exclusion criteria were inability to stand and amputation. This study was approved by the Chang Gung Medical Foundation Institutional Review Board (IRB number 202001820B0A3).

### 2.2. Measures and Instruments

Bioelectrical impedance analysis (BIA) is a secure, quick, radiation-free, and non-invasive method to assess body composition in a community setting [13,21]. BIA is useful in epidemiological and community settings and for clinical diagnosis of sarcopenia [22].

During routine revisits, patient body composition was measured through multifrequency BIA (MF-BIA; InBody 770, Cerritos, CA, USA). The MF-BIA device is a universal, convenient, instantaneous, non-invasive, and highly accurate bioelectrical impedance analyzer. Thus, it is currently the most common for measuring body composition. MF-BIA devices can assess changes in body composition (i.e., fat mass, FFM, and PBF) [23,24,25].

Data on demographic characteristics, BMI (kg/m^2^), body fat mass (BFM; calculated as total weight (kg) minus FFM (kg)), FFM (muscles, bones, organs, and body fluids (kg)), PBF body fat mass/weight × 100% (%)), visceral fat area (VFA, the estimated area of fat surrounding internal organs in the abdomen; a VFA under 100 cm^2^ should be maintained to remain healthy) [26], appendicular skeletal muscle mass (ASM (kg)), and skeletal muscle index (SMI, ASM/height^2^ (kg/m^2^)) were collected for statistical analyses. 

### 2.3. Definitions of Variables

The cutoff points for BMI categories in Taiwan are defined as follows: underweight: <18.5 kg/m^2^, normal weight: 18.5 to <24 kg/m^2^, overweight: ≥24 to <27 kg/m^2^, and obesity: ≥27 kg/m^2^. Low muscle mass was defined using SMI (male: <7.0 kg/m^2^; female: <5.7 kg/m^2^); sarcopenic obesity was defined using SMI and PBF (male: SMI <7.0 kg/m^2^ and PBF ≥25%; female: SMI <5.7 kg/m^2^ and PBF ≥30%); and obesity was defined using BMI (≥27 kg/m^2^) or PBF (male: ≥25%; female: ≥30%). Previous research was referenced for the definitions of the study variables [27].

### 2.4. Statistical Analyses

SPSS 21 (IBM Corp., Armonk, NY, USA) was used to perform statistical analysis. Continuous variables are displayed as mean ± standard deviation, and categorical variables are presented in percentages. Analysis was performed in five major parts. First, the participants were divided by sex, and continuous variables were analyzed using independent *t*-tests stratified by sex (Table 1). Second, the participants were divided by sex and age, and the ratios of low muscle mass and sarcopenic obesity were calculated (Table 2). Third, obesity was defined using PBF and BMI, and obesity prevalence was calculated (Figure 1). Fourth, the prevalence of body composition anomalies in different BMI groups was calculated (Table 3). Finally, one-way ANOVA and independent *t*-tests were used to calculate differences in body composition distribution by age and sex (Table 4). Differences were considered statistically significant when *p* < 0.05.

## 3. Results

### 3.1. Demographics of Participants

This study included 2404 patients, with an average age of 63.2 ± 12.9 years and T2DM disease duration of 12.5 ± 7.9 years. The study sample included more men (53%), and the average glycated hemoglobin was 7.3% ± 1.1%. Regarding body composition differences between sexes, female participants had higher mean values of BFM, PBF, and VFA and lower mean values of FFM, ASM, and SMI compared with men (Table 1).

### 3.2. Prevalence of Low Muscle Mass and Sarcopenic Obesity in Different Age and Sex Groups

As suggested by the results in Table 2, the prevalence of low muscle mass and sarcopenic obesity was 28.0% and 18.7%, respectively, and it increased with age. The prevalence of low muscle mass and sarcopenic obesity was higher among women than among men.

### 3.3. Obesity Prevalence

As indicated by Figure 1, overall obesity prevalence when PBF was used was 71.5%, which was higher than that when BMI was used (32.4%), and the female group had a higher prevalence than the male group (82.6% and 61.6%, respectively).

### 3.4. Prevalence of Body Composition Anomalies in Different BMI Groups

Table 3 displays the prevalence of low muscle mass (55.6%), sarcopenic obesity (34.8%), and obesity (47.8%, when PBF was used) of people with normal BMI (18.5 ≤ BMI < 24 kg/m^2^).

### 3.5. Body Composition by Age and Sex

As suggested by the results in Table 4, for both men and women, body weight (BW), BFM, FFM, ASM, and SMI all decreased with age.

## 4. Discussion

This study demonstrated that the prevalence of low muscle mass and sarcopenic obesity was 28% and 18.7%, respectively. Overall, the prevalence among women was higher than that among men, and it increased with age. The prevalence in this study was higher than the prevalence of 7–9% reported by Kim et al. but lower than the prevalence of 30–34% reported by Yang et al. These differences were determined to be caused by differences in measurement equipment, definitions of low muscle mass, and study groups. However, in all three studies, the prevalence of low muscle mass and sarcopenic obesity among older adults was higher than that among other age groups [28,29]. Research has indicated that sarcopenia and sarcopenic obesity cause a decline in body function, resulting in frailty, falls, disability, and death [30]. Moreover, decreased muscle mass increases the risk of metabolic syndrome [31]. In addition, muscle mass is a crucial factor influencing mortality [9]. Low muscle mass predicts all-cause mortality more accurately than low BMI does. Furthermore, sufficient muscle mass is closely correlated with older adult health and quality of life [32].

This study determined that the prevalence of low muscle mass increased with age, as did that of sarcopenic obesity. The prevalence of both low muscle mass and sarcopenic obesity was highest in the normal BMI group (18.5 ≤ BMI < 24 kg/m^2^). This suggests that BMI cannot accurately distinguish between muscle mass and fat mass, and it is clearly not a precise prediction factor. Thus, using this index to diagnose obesity may result in underestimation of the prevalence of obesity.

One meta-analysis indicated that over half the participants had high PBF, but the diagnosis of obesity could not be established through screening BMI [33]. In addition, the BMI cutoff point cannot reveal visceral fat and central obesity problems [34]. Although BMI provides a convenient and fast standard for identifying obesity, body composition factors, such as fat mass and distribution, also influence health [11]. Chang et al. demonstrated that a low-BMI-to-high-body-fat ratio is more prevalent among Taiwanese people than among Western people. They recommended modifying the current obesity standard of BMI to ≥25 kg/m^2^ to make it more suitable for the Taiwanese population [35]. Based on the current study results, fat mass and muscle mass should be considered together when discussing the effects of BMI on health.

BW, BFM, FFM, ASM, and SMI decreased with age. This is consistent with previous research results. Muscle mass decreases with age, whereas abdominal fat and FFM increase with age [36]. Muscle fibers in limbs decrease after the age of 40 years. Muscle mass starts to decrease from the age of 50 years, and the decrease rate accelerates after the age of 70 years. The subcutaneous fat of older adults decreases, but their visceral fat increases with age [37]. A South Korean cohort study targeting the adult population determined that relative muscle mass was negatively correlated with T2DM [38]. Skeletal muscles account for a large ratio in body composition, and they are crucial for glucose consumption, storage, and metabolism [39].

The present study uncovered that young female adults (18 to <35 years) had higher mean values of VFA than did other age groups. Young women should pay attention to the subsequent impact of VFA because VFA is closely related to metabolic syndrome in young individuals [13,16]. The accumulation of abdominal subcutaneous adipose tissue (SAT) (especially deep-layer SAT) is a strong predictor of insulin resistance and liver-specific insulin resistance [40]. The results of the famous Framingham Study indicated that abdominal obesity increases the risk of cardiovascular diseases. Fat tissue on the mesentery releases multiple inflammatory substances, such as IL-6. Such substances not only have adverse effects on vascular endothelial cells but also increase insulin resistance. Both are unfavorable for stable blood sugar control [40]. Another study reported that visceral fat gain might induce β-cell failure in compensation for insulin resistance, increasing the risk of diabetes [41].

In addition, research has demonstrated that when patients with T2DM restrict their diet intake under the influence of chronic diseases, this eventually causes sarcopenia and frailty [32]. In the current study, caregivers are recommended to provide health education information related to obesity and insufficient muscle mass to patients with abnormal PBF, BMI, or skeletal muscle weight. Such information should include exercise and nutrition for decreasing body fat and increasing skeletal muscle. Muscle strength and endurance should be improved to reduce the health risk caused by obesity and insufficient muscle mass.

### Limitations

This study had several limitations. First, lifestyle factors, such as nutrition status, diet behavior, and physical exercise were not included in the analysis, which may result in residual confounding. Second, this was a cross-sectional study, and changes in body composition and their effects were not traced in the long term. Thus, additional studies are required for verification. Third, for patients with complications from other chronic diseases, physiological and blood biochemical indices are required for assessment to clarify the changes in body composition caused by the diseases. Fourth, the universality of the study may be limited due to the specific population under analysis. Most participants were adults aged over 35 years. Thus, caution is required when extending the findings to younger populations. Lastly, although InBody measurement has advantages, previous studies have described its limitations, including that the method underestimates body fat, fat mass, and PBF and overestimates FFM compared to the dual-energy X-ray absorptiometry (DXA); these deviations are largest among normal-weight people and decrease with increasing BMI [23,42]. Thus, healthcare providers must understand the limitations of body composition assessment methods.

## 5. Conclusions

The results revealed that the overall PBF of individuals living with T2DM was too high. Furthermore, the prevalence of low muscle mass and sarcopenic obesity was high in older adults and people with normal BMI. Using BMI to assess obesity and insufficient muscle mass underestimates the prevalence of obesity and neglects the problems of sarcopenia and high body fat in people with normal BMI. Clinical practitioners are recommended to improve screening for sarcopenia and PBF for older patients and people with normal BMI. Thus, their health can be improved through nutrition and exercise interventions.

## Figures and Tables

**Figure 1 ijerph-18-04421-f001:**
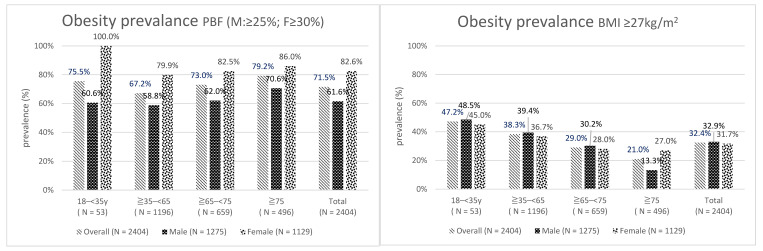
Obesity prevalence.

**Table 1 ijerph-18-04421-t001:** Demographics of participants.

Variable	Overall *N* = 2404 *N* (%)/Mean ± *SD*	Male *N* = 1275 (53.0%) *N* (%)/Mean ± *SD*	Female *N* = 1129 (47.0%) *N* (%)/Mean ± *SD*
Age	63.2 ± 12.9	60.6 ± 12.4	69.9 ± 11.6
18–<35	53 (2.2%)	33 (2.6%)	20 (1.8%)
≧35–<65	1196 (49.8%)	719 (56.4%)	477 (42.2%)
≧65–<75	659 (27.4%)	305 (23.9%)	354 (31.4%)
≧75	496 (20.6%)	218 (17.1%)	278 (24.6%)
Disease duration	12.5 ± 7.9	12.1 ± 7.9	12.9 ± 7.9
A1C	7.3 ± 1.1	7.3 ± 1.1	7.1 ± 1.1
Height (m)	1.60 ± 0.09	1.66 ± 0.07	1.53 ± 0.06
BW (kg)	65.8 ± 14.0	71.4 ± 13.7	59.6 ± 11.6
BMI (kg/m^2^)	26.9 ± 4.1	25.7 ± 4.2	25.4 ± 4.4
<18.5	44 (1.8%)	23 (1.8%)	21 (1.9%)
≧18.5–<24	890 (37.0%)	426 (33.4%)	464 (41.1%)
≧24–<27	692 (28.8%)	406 (31.8%)	286 (25.3%)
≧27	778 (32.4%)	420 (32.9%)	358 (31.7%)
BFM (kg)	20.9 ± 8.0	19.8 ± 8.0	22.2 ± 7.8
FFM (kg)	44.9 ± 9.7	51.6 ± 7.8	37.4 ± 5.2
PBF (%)	31.4 ± 8.1	27.0 ± 6.6	36.3 ± 6.6
VFA	102.3 ± 42.6	91.0 ± 38.2	115.1 ± 43.7
ASM (kg)	18.2 ± 4.8	21.5 ± 3.7	14.5 ± 2.7
SMI (kg/m^2^)	7.0 ± 1.2	7.7 ± 0.9	6.1 ± 0.9

Note: A1C: hemoglobin A1C; body fat mass (BFM) = total weight minus fat-free mass (FFM); FFM = weight of muscles + bones + organs + body fluids; percentage body fat (PBF) = BFM/body weight (BW) × 100%; visceral fat area (VFA): the estimated area of fat surrounding internal organs in the abdomen; Appendicular skeletal muscle mass (ASM); skeletal muscle index (SMI) = ASM/height (m)^2^.

**Table 2 ijerph-18-04421-t002:** Prevalence of low muscle mass and sarcopenic obesity among participants of different age groups.

Variable	Overall (*N* = 2404)		Male (*N* = 1275)		Female (*N* = 1129)	
	Low Muscle Mass	Sarcopenic Obesity	Low Muscle Mass	Sarcopenic Obesity	Low Muscle Mass	Sarcopenic Obesity
Age						
18<35	5 (9.4%)	3 (5.7%)	2 (6.1%)	0 (0.0%)	3 (15.0%)	3 (15.0%)
≧35–<65	195 (16.3%)	112 (9.4%)	88 (12.2%)	36 (5.0%)	107 (22.4%)	76 (15.9%)
≧65–<75	204 (31.0%)	133 (20.2%)	81 (26.6%)	39 (12.8%)	123 (34.7%)	94 (26.6%)
≧75	269 (54.2%)	201 (40.5%)	118 (54.1%)	81 (37.2%)	151 (54.3%)	120 (43.2%)
Total	673 (28.0%)	449 (18.7%)	289 (22.7%)	156 (12.2%)	384 (34.0%)	293 (26.0%)

Note: Low muscle mass is defined with skeletal muscle index (SMI) (male: <7.0 kg/m^2^; female: <5.7 kg/m^2^); sarcopenic obesity is defined with SMI and percentage body fat (PBF) (male: SMI < 7.0 kg/m^2^ and PBF ≥ 25%; female: SMI < 5.7 kg/m^2^ and PBF ≥ 30%).

**Table 3 ijerph-18-04421-t003:** Prevalence of body composition anomalies in different BMI groups.

	Category	Low Muscle Mass			Sarcopenic Obesity			Obesity		
BMI		Overall (*N* = 2404)	Male (*N* = 1275)	Female (*N* = 1129)	Overall (*N* = 2404)	Male (*N* = 1275)	Female (*N* = 1129)	Overall (*N* = 2404)	Male (*N* = 1275)	Female (*N* = 1129)
<18.5		42 (95.5%)	22 (95.7%)	20 (95.2%)	3 (6.8%)	0 (0.0%)	3 (14.3%)	3 (6.8%)	0 (0.0%)	3 (14.3%)
≧18.5–<24		495 (55.6%)	216 (50.7%)	279 (60.1%)	310 (34.8%)	105 (24.6%)	205 (44.2%)	425 (47.8%)	129 (30.3%)	296 (63.8%)
≧24–<27		114 (16.5%)	43 (10.6%)	71 (24.8%)	114 (16.5%)	43 (10.6%)	71 (24.8%)	537 (77.6%)	262 (64.5%)	275 (96.2%)
≧27		22 (2.8%)	8 (1.9%)	14 (3.9%)	22 (2.8%)	8 (1.9%)	14 (3.9%)	753 (96.8%)	395 (94.0%)	358 (100.0%)
Total		673 (28.0%)	289 (22.7%)	384 (34.0%)	449 (18.7%)	156 (12.2%)	293 (26.0%)	1718 (71.5%)	786 (61.6%)	932 (82.6%)

**Table 4 ijerph-18-04421-t004:** Body composition of different age groups and sexes.

Variable	18–<35		≧35–<65		≧65–<75		≧75	
	Male (*N* = 33)	Female (*N* = 20)	Male (*N* = 719)	Female (*N* = 477)	Male (*N* = 305)	Female (*N* = 354)	Male (*N* = 218)	Female (*N* = 278)
Height (m)	1.73 ± 0.06	1.60 ± 0.05	1.68 ± 0.06 ***	1.55 ± 0.06 **	1.64 ± 0.05 ***	1.52 ± 0.05	1.63 ± 0.06 ***	1.50 ± 0.06
BW (kg)	85.8 ± 21.3	68.1 ± 13.3	74.6 ± 13.8 ***	62.6 ± 12.7	68.2 ± 11.1 ***	57.9 ± 9.7 ***	63.2 ± 9.0 ***	55.8 ± 10.0 ***
BMI (kg/m^2^)	28.5 ± 6.7	26.6 ± 4.6	26.3 ± 4.3 *	26.1 ± 4.9	25.3 ± 3.7 ***	25.0 ± 3.9	23.9 ± 3.1 ***	24.8 ± 3.9
BFM (kg)	25.4 ± 14.3	26.3 ± 8.5	20.4 ± 8.2 **	23.2 ± 8.7	19.0 ± 7.2 ***	21.4 ± 7.0 *	18.1 ± 5.9 ***	21.1 ± 6.7 *
FFM (kg)	60.4 ± 9.1	41.8 ± 6.1	54.1 ± 7.5 ***	39.4 ± 5.5	49.2 ± 5.7 ***	36.5 ± 4.1 ***	45.1 ± 5.0 ***	34.7 ± 4.5 ***
PBF (%)	27.7 ± 9.3	37.9 ± 5.7	26.6 ± 6.6	36.0 ± 6.9	27.0 ± 6.6	36.1 ± 6.5	28.1 ± 6.1	37.0 ± 6.3
VFA (cm^2^)	108.2 ± 62.3	127.3 ± 43.4	91.8 ± 39.2	115.5 ± 46.1	88.3 ± 36.3 *	112.4 ± 42.7	89.6 ± 31.8 *	117.0 ± 40.7
ASM (kg)	25.4 ± 3.9	16.8 ± 2.8	22.7 ± 3.6 ***	15.6 ± 2.7	20.4 ± 2.8 ***	14.0 ± 2.1 ***	18.5 ± 2.6 ***	13.0 ± 2.4 ***
SMI (kg/m^2^)	8.4 ± 1.0	6.5 ± 0.8	8.0 ± 0.9 *	6.4 ± 0.9	7.6 ± 0.8 ***	6.0 ± 0.7*	7.0 ± 0.8 ***	5.7 ± 0.8 ***

Note: All data are means ± standard deviation; for the significance, the 18 to 34 age group is used as a reference for comparison between the groups; BMI, body mass index; BFM, body fat mass; FFM, free fat mass; PBF, percentage body fat; VFA, visceral fat area; ASM, appendicular skeletal muscle mass; SMI, skeletal muscle index; * *p* < 0.05, ** *p* < 0.01, *** *p* < 0.001.

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
