# Peer review of "Association of Body Composition with Type 2 Diabetes: A Retrospective Chart Review Study"

_ijerph, 2021, doi:10.3390/ijerph18094421_

Round 1

Reviewer 1 Report

I appreciate the opportunity to review this interesting paper on body composition in Type 2 Diabetes patients. Lin et al., in a retrospective study, analyzed the body composition of subjects with Type 2 Diabetes measured with a Multi-Frequency Bioelectrical Impedance analysis (MF-BIA; InBody 19 770). They found that low muscle mass and sarcopenic obesity were high in older adults and normal BMI subjects. They concluded that using BMI to assess obesity underestimates obesity prevalence and leads to sarcopenia unnoticed in normal BMI subjects.

I commend the authors for several strengths of their work, including addressing an important and timely question.

Relying only on BMI can lead to false conclusions. Using more reliable methods of measuring body composition is especially important in older individuals at risk of sarcopenia.

 The subject is in the range of the journal, and the manuscript is of clinical relevance.

Considering these strengths, though, I found some areas where I would have appreciated greater clarity as I read the manuscript.

  • The first doubt concerns the method itself, i.e., the measurement of composition using measured with a Multi-78 Frequency Bioelectrical Impedance analysis (MF-BIA; InBody 770). The authors state that "it is currently the most commonly used clinical method for measuring body composition." While the method actually works when we are to assess body composition changes over time in an individual (e.g., in response to diet, training, or illness), it has severe limitations [1-4]. I want to draw the authors' attention to a recently published paper by Lahav et al. [4]. They found that InBody underestimated body fat (BF%) assessment and that these deviations were largest among normal-weight people and decreased with increasing BMI group. On the other hand, Antonio et al [3] showed that this measurement underestimated fat mass and percentage body fat and overestimated fat-free mass. This method is also not optimal for measuring skeletal muscle tissue [2]. Authors should review the literature regarding the methods used and devote a few paragraphs to this controversy. These reservations should also appear in Limitations. My objections do not call into question the authors' conclusions, but they should give readers a clear picture and allow them to draw their conclusions.
  • Study participants had visceral fat levels (VFL) measured, but VFL was not included in the analysis. And yet (as the authors themselves mention in the Introduction), visceral adipose tissue, to a greater extent than total fat, has pathological consequences. The assessment of visceral adiposity is of great significance for preventing metabolic disorders, especially in non-obese individuals [5,6].
  • Another problem with this manuscript is that the authors do not demonstrate their field mastery in their introduction and discussion sections. Too few studies are cited. I am not left fully convinced of the novelty of this work.
  • The title does not fully reflect the paper's content, which is primarily devoted to comparing PBF and BMI as indices from obesity.
  • Tables are not well described. They should explain data without the reader needing to go and dive into the text. This is especially true of table 3. It requires some effort on the part of the reader before understanding what it is about.

  1. Jeon, K.C.; Kim, S.-Y.; Jiang, F.L.; Chung, S.; Ambegaonkar, J.P.; Park, J.-H.; Kim, Y.-J.; Kim, C.-H. Prediction Equations of the Multifrequency Standing and Supine Bioimpedance for Appendicular Skeletal Muscle Mass in Korean Older People. International Journal of Environmental Research and Public Health 2020, 17, 5847.
  2. Brewer, G.J.; Blue, M.N.M.; Hirsch, K.R.; Peterjohn, A.M.; Smith-Ryan, A.E. Appendicular Body Composition Analysis: Validity of Bioelectrical Impedance Analysis Compared With Dual-Energy X-Ray Absorptiometry in Division I College Athletes. The Journal of Strength & Conditioning Research 2019, 33, 2920-2925, doi:10.1519/jsc.0000000000003374.
  3. Antonio, J.; Kenyon, M.; Ellerbroek, A.; Carson, C.; Burgess, V.; Tyler-Palmer, D.; Mike, J.; Roberts, J.; Angeli, G.; Peacock, C. Comparison of Dual-Energy X-ray Absorptiometry (DXA) Versus a Multi-Frequency Bioelectrical Impedance (InBody 770) Device for Body Composition Assessment after a 4-Week Hypoenergetic Diet. Journal of Functional Morphology and Kinesiology 2019, 4, 23.
  4. Lahav, Y.; Goldstein, N.; Gepner, Y. Comparison of body composition assessment across body mass index categories by two multifrequency bioelectrical impedance analysis devices and dual-energy X-ray absorptiometry in clinical settings. European Journal of Clinical Nutrition 2021, 10.1038/s41430-020-00839-5, doi:10.1038/s41430-020-00839-5.
  5. Lee, Y.-C.; Lee, Y.-H.; Chuang, P.-N.; Kuo, C.-S.; Lu, C.-W.; Yang, K.-C. The utility of visceral fat level measured by bioelectrical impedance analysis in predicting metabolic syndrome. Obesity Research & Clinical Practice 2020, 14, 519-523.
  6. Liu, X.Z.; Chen, D.S.; Xu, X.; Li, H.H.; Liu, L.Y.; Zhou, L.; Fan, J. Longitudinal associations between metabolic score for visceral fat and hyperuricemia in non-obese adults. Nutrition, Metabolism and Cardiovascular Diseases 2020, 30, 1751-1757.

Author Response

Dear reviewer:

          Please find the attachment with our replies.

Reviewer 2 Report

 Association of Body Composition with Type 2 Diabetes: A Retrospective Chart Review Study

Abstract

Line 20. Please after point must be Capital letter: 770). Subjects

The abstract explains well the research objectives and helps the future reader to get into the paper by giving a preview of what has been done and what the main results and general conclusion are.

Excellent

Introduction

Line 43. Please, include one space between words an numbers: US$ 727 billion.

On line 43, where the first reference appears, it is included after the point. I understand that this reference is to the information that precedes it and therefore the number in superscript should appear first, followed by the dot. Please correct this throughout the text.

In this introduction we will add some approaches to the possible differences between having or not having diabetes for which I recommend this reference (Palomo, C. y Denman, C. A. (2019) «Actividad física en adultos con y sin diabetes en México (ENSANUT MC-2016)», Revista Iberoamericana de Ciencias de la Actividad Física y el Deporte, 8(3), pp. 13-28. doi: 10.24310/riccafd.2019.v8i3.5789.) or with some exercise practices and their influence, please include this quote (Petrovic, M.; Ruiz-Montero, P. Transformación antropológica En Mayores Con Diabetes Tipo II a través Del Ejercicio: Pilates Y Ejercicio aeróbico. RICCAFD 20154, 1-5.).

  1. Materials and Methods

Please advise whether participants consented to participate in the study by signing the informed consent form.

2.2. Measures and instruments

We don't know if the participants prior to being tested with INBODY followed the manufacturer's protocol: no eating, drinking, intense exercise or showering in the two hours prior to the test. The alteration of this protocol significantly alters the assessment due to the modification of the body's hydration which is the main transmitter of the electrical current transmitted by the INBODY.

You should consider at the end of the paper as one of the limitations of this study not to have contemplated the practice of physical exercise. A simple IPAQ-7 questionnaire would provide excellent information and explain the variables that include muscle as a classification variable.

  1. Results

Please justify why you have used these age groups (include references).

Please remember what A1C and VFL mean at the bottom of table 1.

On lines 122-123 it states the following “Table 2 presents the prevalence of low muscle mass and sarcopenic obesity of patients in different age groups”. Please include references to justify these cut-off points.

In the title of table 2 the N should be deleted as it explains, very well, that it is for the different age groups and therefore confuses the reader who will believe that each group has 2404 subjects.

In this journal the decimal separator is com and not dot. Please check this in all tables and text.

In table 2 the parentheses are sometimes separated by a space after the preceding digit and sometimes not. Please correct this.

Table 3, 4 and 5. Sames comments than table 2. Please correct.

  1. Discussion

Without a doubt, this is the best section of the article that I potentially quoted in the future.

It gathers well the information from the introduction and compares well with its results.

I would just include at the end some practical recommendations for future professionals dealing with people with diabetes type 2.

Author Response

(The authors gave the same response as above.)

Reviewer 3 Report

A well done paper. I have no major comments. A few minor comments to consider:

  1. Was the diabetic clinic population representative of Taiwanese diabetic individuals?
  2. Given the ease of bioimpedance measurement, should it be included in the vital signs when seeing patients?
  3. Is there an historical cohort against which to compare your results to those of a few decades ago? A comparison may put your results into perspective.
  4. Line 211- "too cautious about diet" - what does that mean?
  5. The cut points for sarcopenia and BMI are European. Should not Asian measures be used?

Author Response

(The authors gave the same response as above.)

Reviewer 4 Report

Summary:

In this paper the authors present their retrospective chart review study results that were conducted to demonstrate that BMI is not the only index used to assess obesity, and in addition the body composition, the body fat distribution and the scheletal muscle mass play a more important role in the pathophysiology of disease such as diabetes mellitus.

However,  the work is original, the design of the study is appropriate, and the authors have analyzed an impressive amount of data, the work has no novelty value. Moreover, the manuscript requires extensive spelling and grammar correction.

Author Response

Please find the attachment with our replies.

Round 2

Reviewer 4 Report

Summary:

Although the manuscript has been revised by the authors and was corrected according to the recommendations, the revised version of the manuscript requires further improvements. A thorough correction of language and style of the manuscript is strongly recommended.

Please, revise and correct the following sentences:

Line 26: “Subjects with normal BMI the prevalence of low muscle mass (55.6%) and sarcopenic obesity (34.8%) were higher than those of other BMI groups.”

Line 69: “Previous studies indicated higher visceral fat levels is associated with metabolic syndrome (MetS) (Lee et al., 2020) and hyperuricemia in non-obese adults”

Line 76: “…a study indicated T2DM and sarcopenia was a bidirectional relationship”

Line 78: “Beside, compared to sarcopenia or obesity alone, sarcopenic obesity are much greater to health impact and may have synergistic effect with low-grade inflammation to exacerbate insulin resistance, further impairing glucose metabolism”

Line 87: “The case inclusion criteria included being older than 18 years and having completed the body composition measurement.”

Line 101: “….can precise predict changes in body composition”

Line 105 “….total weight of muscles bones,….”

Line 111: “The definition of variables”

Line 117:  “Previous research was referenced for the definitions of variables”

Line 117: “female: 30%”. Please, present the data as accurately as for males.

Line 132: “Demographic of the study subjects”

Line 137: “….compared their counterpart”

Line 139: “Demographic of the study subjects.”

Table 1: Height(m) and SMI(kg/m2). Spacing is required.

Line 148: “The prevalence of people aged over 75 years (54.2%) was higher than that of other age groups.”

Line 151: “The prevalence of people aged over 75 years (54.2%) was higher than that of other age groups.”

Line 167: “ Figure 1. Obesity prevalenc.”

Line 175: “Table 4 is the body composition of different age groups and sexes.”

Line 177: “PBF increased with age for people aged over 65 years,….” In the table, the authors present data for patients over 75 years of age, not those over 65 years of age. Moreover, the value of PBF increases only in case of males (Male: 28.1±6.1 Female: 37.0±6.3) compared to the values of other age groups: Male: 27.7±9.3 Female: 37.9±5.7; Male: 26.6±6.6 Female: 36.0±6.9 and Male: 27.0±6.6 Female: 36.1±6.5.

Line 230: “Present results showed VFA did not exhibit particular trends.”

Line 230: “However, it is note-worthy that young female adults have higher mean of VFA, previous study indicated visceral adipose tissue, to a greater extent than total fat, has pathological consequences.”

Line 240: “Previous study suggest that visceral fat gain may induce ß-cell failure in compensation for insulin resistance, resulting in diabetes regardless of obesity level”

Line 251: “First, lifestyles were not explored in depth, such as nutrition status, diet behaviors and physical exercise, the differences in lifestyles still had randomly distributed characteristics, we suggested to apply simple questionnaire to assess the lifestyles in further study.”

Line 259: “Thus, assessments of BW and body composition should consider such effects to avoid misjudgment.”

Line 267: “In addition to, this method….”

Line 269: “Thus, it would behoove investigators to understand the limitations of this method for body composition assessment methods.”

Line 275: “Clinical practitioners are recommended to improve the screening of sarcopenia and PBF for older patients and people with normal BMI.

Line 284: “…were contributed to acquisition data;…”
